# CELAD: Compositional Evaluation for Logical Anomaly Detection

## Abstract

Anomaly detection (AD) has attracted significant research interest and now achieves near-perfect performance on most existing benchmarks. However, the majority of prior work has focused on detecting *structural* anomalies, where anomalies manifest as localized defective regions. Recently, *logical* anomaly detection (LAD) has emerged, extending AD to cases where violations occur at the level of compositional or relational rules rather than individual regions—a setting particularly relevant for industrial inspection. Despite its importance, LAD research remains hindered by limited dedicated datasets, raising concerns about the generalization ability of current methods. We address this gap with two contributions. First, we introduce CELAD, a new benchmark designed to test compositional understanding in LAD. CELAD features greater variation in both normal and anomalous samples, along with more intricate anomaly types, resulting in a substantial performance drop in state-of-the-art methods. Second, we propose ROMAD, a simple yet effective framework that leverages DETR, an object detector with strong relational modeling, to construct rich object embeddings. ROMAD computes anomaly scores via a training-free matching pipeline and requires only a small number of annotated samples. Extensive experiments show that ROMAD achieves a new state of the art on CELAD, outperforming the next-best method by nearly 15% while maintaining competitive performance on existing datasets. In few-shot regimes, ROMAD further delivers the strongest average results across both CELAD and prior benchmarks, demonstrating its ability to generalize beyond memorization and capture the underlying logical rules. Code and data are available at: `https://github.com/neutral-coder-737/Home-Page`.

## 1 Introduction

Anomaly detection (AD) is a fundamental problem in computer vision with broad applications in industrial and medical domains. Since anomalous samples are rare, diverse, and costly to collect, unsupervised AD has become the most practical and dominant branch of the field. In this context, models are given a training set of solely normal samples to learn their distribution. During inference, any sample that deviates from the established distribution of normal samples should be recognized as anomalous. Much of the progress in the field of AD has been driven by benchmarks such as MVTec AD (Bergmann et al., 2019) and Visa (Zou et al., 2022), which focus primarily on structural anomalies. These types of anomalies include defects like dents or scratches that disrupt the uniformity of the image, making it easy to isolate the affected area. As a result of these rapid advances, there are various well established methods, many of which now achieve near-perfect detection performance.

However, the success of structural AD has highlighted the limitations of current benchmarks and motivated the emergence of logical anomaly detection (LAD), first introduced with the MVTec LOCO AD dataset (Bergmann et al., 2022). Unlike structural anomalies, logical anomalies involve more intricate relations and extend beyond individually contained defects. In this type of anomalies, while no single object of the image is flawed on its own, the overall composition of these elements and the underlying logical rule is violated. This paradigm is particularly relevant for industrial quality inspection, where products often adhere to strict logical constraints. Despite recent progress, existing LAD research still has two key shortcomings. First, the true practical applications of LAD are substantially richer and more complex than what the existing benchmarks cover, raising concerns of overfitting and limited generalization. Second, most real-world scenarios, lack

the capacity for acquiring a comprehensive set of training samples, making *few-shot LAD* especially important. Moreover, reliable LAD systems are expected to capture the underlying logical rules of a scene rather than memorizing all the individual instances. This further emphasizes the need for the performance evaluation of LAD methods under few-shot regimes.

To address these challenges, we introduce **CELAD** (Compositional Evaluation for Logical Anomaly Detection), a new benchmark with a special focus on testing the compositional understanding in AD. CELAD presents several new characteristics: images with a larger number of constituent objects, higher variation in both normal and anomalous samples and more complicated compositional anomalies. We show that state-of-the-art LAD methods, which achieve near-saturated performance on prior benchmarks, suffer drops of over 20% on CELAD.

We further present **ROMAD** (Relation-aware Object Matching for logical Anomaly Detection), a simple yet effective framework for LAD. Our approach leverages the powerful relation-modeling capabilities of DETR (Carion et al., 2020), a transformer-based family of object detectors, to create rich object-level embeddings that capture the semantic, positional and compositional information of each object. With only a small number of annotated samples for fine-tuning the object detector, we construct a lightweight, training-free matching pipeline that computes anomaly scores from the matching distances of these embeddings. Our design achieves balanced performance across both existing and new benchmarks. On average over LOCO and CELAD, it ranks second overall, trailing the best method by only 0.5%. On CELAD, however, it establishes a new state-of-the-art, surpassing the second-best approach by nearly 14.7%. Moreover, in the few-shot setting, our method achieves the best average performance across both LOCO and CELAD.

In summary, our contributions are threefold:

- We introduce CELAD, a new benchmark for LAD with a strong emphasis on compositionality and positional relations. CELAD contains more complex anomalies, requiring models to reason over both semantics and compositions. Experiments on CELAD show a significant performance decline across all baselines, with average drops exceeding 20%.

- We propose ROMAD, a relation-aware object matching method that builds on DETR family of object detectors. ROMAD leverages object-level embeddings enriched with compositional cues, and combines it with a training-free matching pipeline to compute the anomaly scores.

- We demonstrate that ROMAD achieves a more balanced performance across both existing and new benchmarks. It delivers state-of-the-art results on CELAD with a 14.7% margin, while maintaining competitive performance across both datasets. Importantly, ROMAD also maintains the strongest average results across both datasets in the few-shot setting.

## 2 RELATED WORK

**Structural Anomaly Detection.** Conventional approaches have largely focused on *unsupervised* AD and a variety of strategies have been proposed: *Retrieval-based methods* construct a memory bank from feature maps of normal samples and assign anomaly scores by nearest-neighbor search over local patches of the test image (Cohen & Hoshen, 2020; Roth et al., 2022). *Density estimation methods* attempt to model the distribution of normal data explicitly, often under specific assumptions such as multivariate Gaussian distributions (Defard et al., 2021), or through more flexible approaches such as normalizing flows (Rudolph et al., 2022; 2023). The likelihood of a test sample under this distribution is then used as the anomaly score. *Reconstruction-based methods* rely on the assumption that generative models trained exclusively on normal data fail to reconstruct anomalies. At test time, discrepancies between the input and its reconstruction highlight anomalous regions (Zhang et al., 2024b; Mousakhan et al., 2024). *Distillation-based methods* train a student network to mimic a teacher using only normal data. The discrepancy between teacher and student outputs is then used as the anomaly score (Salehi et al., 2021; Deng & Li, 2022).

**Logical Anomaly Detection.** Building on these foundations, recent work has shifted attention toward LAD, which focuses on relational and compositional inconsistencies rather than low-level structural defects. Many LAD approaches are extensions of conventional AD pipelines with architectural or loss-function modifications to better capture long-range dependencies. For instance, the

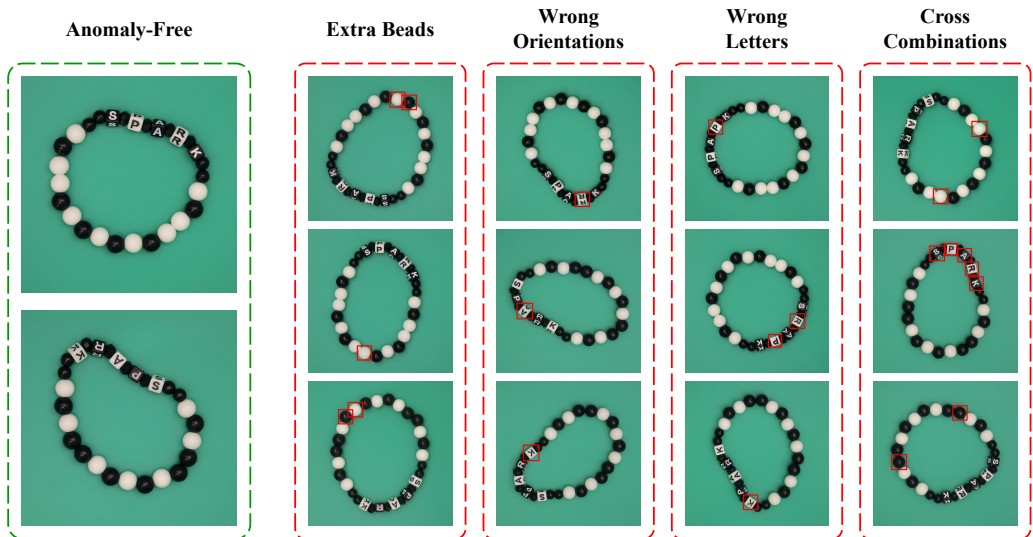

Figure 1: Preview of the twin bracelets class in the proposed CELAD dataset. The images illustrate normal bracelet compositions alongside various anomaly types. Anomalous regions are highlighted with bounding box annotations. For images with multiple valid anomaly labels, only one annotation is visualized for clarity.

use of *reconstruction–distillation hybrids* with adaptations for long-range understanding have been widely adopted (Bergmann et al., 2022; Batzner et al., 2024; Sugawara & Imamura, 2024; Zhang et al., 2024a). Some other works use *deep feature reconstruction*, where bottleneck compression is used to filter out anomalous features (Guo et al., 2023; Patra & Taieb, 2024). *Density-based methods* such as SINBAD, model each sample as a distribution of its elements, using fixed features and then compute anomaly scores via a simple density estimation method (Cohen et al., 2023). *Segmentation-based methods* first decompose the image into constituent components and then perform matching to compare the object classes of the test image against those of its nearest normal neighbors. By constructing pixel-level segmentation maps, these methods are particularly effective at capturing the metrological features of objects (Kim et al., 2024; Hsieh & Lai, 2024; Liu et al., 2023; Peng et al., 2025). Other lines of work include methods leveraging the reasoning capabilities of large language models or program synthesis to detect logical violations (Zhang et al., 2025; 2024c). While these methods are attractive for their accuracy and explainability, they remain impractical for real-world deployments due to their high computational overhead and limited real-time applicability. Our approach is most closely related to the segmentation-based family, with the distinction that (i) we use object detection, which serves as a lighter base task compared to segmentation, (ii) we perform matching at the object level with a focus on modeling compositional context, and (iii) our framework is simpler, leveraging existing vision foundation models rather than custom architectures.

**Discussion.**    Notably, most LAD pipelines address logical and structural anomalies through separate branches, effectively ensembling models to handle both types. This trend, combined with the clear performance gap between LAD and structural AD, highlights that LAD remains underexplored and warrants its own dedicated line of research. To this end, our work exclusively targets LAD: we introduce a new benchmark that explicitly emphasizes compositional relations, and a lightweight object-matching framework aimed at advancing progress in this emerging direction.

## 3    THE CELAD DATASET

### 3.1    MOTIVATION

Recent advancements in LAD have been significantly enabled by the introduction of the MVTec LOCO AD dataset (Bergmann et al., 2022). Encompassing five different object classes, each having

a specific set of logical rules, this dataset has served as the main benchmark for the task of LAD. Despite the challenging nature of these logical anomalies, current state-of-the-art methods, have achieved near-perfect performance on this benchmark. However, the complex nature of real-world logical anomalies leads us to question the generalization ability of existing methods, suggesting that the problem of generalizable LAD is far from solved. This motivates the need for a more challenging benchmark to drive further research in this field. Compared to the previous data, CELAD features a significantly higher degree of variation in both normal and anomalous samples, introducing more complex compositional relationships. By providing high intra-class variation for normal data and a wide variety of anomalies with subtle differences, CELAD serves as a well-suited complementary benchmark to existing datasets. As we show in our experimental evaluation, the performance of prior methods on CELAD drops drastically, highlighting their lack of generalization to more complicated scenarios.

## 3.2 DATASET DESCRIPTION

The CELAD dataset contains images of beaded bracelets, exclusively covering the logical types of anomalies. Anomaly-free samples are defined by two types of bracelets, which exhibit subtle distinctions in the type and the arrangement of their constituent beads. Each bracelet type is defined by letter beads spelling "SPARK" and a unique combination of letter-bead colors and black–white bead arrangements. Any sample deviating from these two precise types is considered anomalous. Figure 1 provides a visual preview of the dataset, showcasing examples of both normal and anomalous samples. Appendix A provides a detailed description of CELAD's logical rules as well as a preview of its pixel-precise annotations. The CELAD dataset comprises 530 normal samples and a total of 220 anomalous samples across five distinct anomaly categories. Compared to each class of LOCO, CELAD offers almost twice as many anomalous test samples, substantially more anomaly types, a larger number and greater variety of constituent objects per image, and overall more challenging anomalies. Table 4 in Appendix A provides a detailed overview of the dataset statistics, including the number of samples and subtypes for each anomalous category. All images where acquired using a 4000×4000 high-resolution camera. Both images and pixel-precise annotations are provided.

## 3.3 EVALUATION PROTOCOL AND METRICS

**Anomaly Detection.** Regarding image-level evaluation, we adopt standard metrics from the field, including Area Under the Receiver Operating Characteristic Curve (AUROC), Average Precision (AP), and F1-score at the optimal threshold (F1-max). These metrics provide a comprehensive measure of a model's ability to distinguish between normal and anomalous samples.

**Anomaly Segmentation.** Pixel-level evaluation for LAD presents a challenge due to the potential for multiple valid ground truth annotations. The LOCO dataset aimed to address this ambiguity with saturated Per-Region Overlap (sPRO), an extended version of Per-Region Overlap (PRO). In this protocol, the union of all candidate regions is selects as the ground truth and the performance scores are saturated once the predicted region's overlap with the ground truth exceeds a predefined threshold. While sPRO lacks the ability to capture the true shape of the anomalous region, it is required for the evaluation of LOCO due to the uncountable number of pixel-level ground truths. In contrast to LOCO, the locations of anomalous regions in CELAD are fixed, and the number of possible ground truths for any given anomalous image is finite. This characteristic allows for a more basic pixel-level evaluation protocol for CELAD: among all possible choices of ground truths for a given anomalous image, the one that has the highest Intersection over Union (IoU) with the model's predicted anomaly map is selected. Performance is then evaluated using standard pixel-level metrics commonly used in structural AD, such as PRO and pixel-level AUROC.

## 4 METHOD

### 4.1 PRELIMINARY: DETR

Our framework is heavily inspired by DETR (Carion et al., 2020), which marked a major shift in object detection by introducing a transformer-based encoder–decoder architecture along with the concept of object queries. In DETR, a fixed set of learnable object queries is passed through the

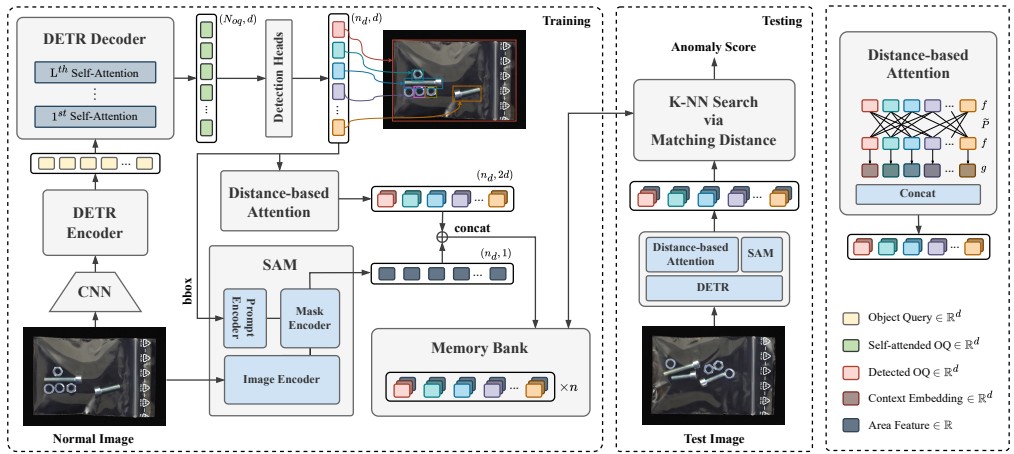

Figure 2: Overview of ROMAD. After fine-tuning the object detector, detected object queries are extracted and enriched with distance-based attention and SAM-derived area features to form representative embeddings of normal images, stored in a memory bank. At test time, embeddings from a query image are compared against the memory bank via bipartite matching, and the resulting distance is used as the anomaly score.

transformer decoder, where self-attention (Vaswani et al., 2017) allows each query to interact with all others. This mechanism naturally enables relation modeling: the embedding of each object is contextualized by attending to other potentially relevant objects in the scene. Unlike traditional detection pipelines that rely on hand-crafted components such as anchors or post-processing heuristics, DETR formulates detection as a direct set prediction problem. It employs bipartite matching to align predictions with ground-truth objects, enabling a fully end-to-end training pipeline. These design choices give DETR a flexible formulation that excels not only at detecting individual objects but also at capturing their interactions. For this reason, DETR is an appealing choice for tasks like LAD, where the understanding of inter-object relations is essential. The contextualized object query embeddings encode positional, semantic, and compositional information of each object—respectively guided by the bounding-box regression head, the classification head, and the self-attention mechanism. Therefore, we propose to use these embeddings as proxies for the objects in a scene, establishing an efficient foundation for object matching.

In practice, we observe that only a small number of annotated data is sufficient to achieve near-perfect detection on AD datasets. Also, since smaller architectures are more aligned with low-data regimes, we adopt RT-DETR (Zhao et al., 2024), a lightweight and efficient variant of the DETR family. RT-DETR mitigates the heavy computational demands typical of transformer-based architectures, while achieving performance and efficiency comparable to state-of-the-art YOLO (Redmon et al., 2016) detectors, making it particularly well-suited for our setting.

## 4.2 ENHANCING OBJECT DETECTION VIA SELF-TRAINING

**Pseudo-label Verification.** An intrinsic property of LAD datasets is that, for anomaly-free samples, the number of objects in each category follows a predefined pattern. This allows us to design a simple verification procedure to score the predicted outputs of the object detector. Concretely, we construct a set of reference histograms $\{h_1^{(r)}, \ldots, h_K^{(r)}\}$ from a small set of manually annotated samples, where $h_{k,c}^{(r)}$ denotes the expected number of objects of category $c$ in reference sample $k$. We then apply the object detector to the training set and compute a histogram $h^{(p)}$ for each predicted image, where $h_c^{(p)}$ is the predicted count for category $c$. We then measure the distance between $h^{(p)}$ and each reference histogram using the $L_2$ distance. A verification score is then assigned as the minimum distance across all reference histograms. this is defined as

$$d_{\text{ver}}(h^{(p)}) = \min_{k=1,\ldots,K} \sqrt{\sum_{c\in\mathcal{C}} \left(h_c^{(p)} - h_{k,c}^{(r)}\right)^2}, \tag{1}$$

where $\mathcal{C}$ is the set of categories. Finally, a predicted annotation is retained as a pseudo-label if its verification score satisfies $d_{\mathrm{ver}}(h^{(p)}) \leq \tau_{\mathrm{ver}}$, where $\tau_{\mathrm{ver}}$ is a predefined threshold. This score serves as a minimal evaluation metric for detection performance based solely on normal samples, enabling us to filter out inaccurate pseudo-labels and retain only the most consistent ones.

**Self-training.** Prior work in object detection has demonstrated that self-training can significantly enhance fine-tuning, particularly in low-data regimes (Zoph et al., 2020). While conventional supervised fine-tuning already yields satisfactory detection results, we argue that the reliance on manually annotated samples can be further reduced through self-training. We also note that, although DETR eliminates the need for post-processing such as the non-maximum suppression (NMS), applying NMS yields a slight accuracy gain in our low-shot setting; hence, we retain it in our pipeline. Our self-training procedure consists of four steps: (i) a teacher model is trained on a small set of manually labeled samples; (ii) the teacher model generates pseudo-labels on the entire training set; (iii) a verification procedure is applied to filter out incorrect pseudo-labels; and (iv) a student model is trained jointly on both the verified pseudo-labels and the manual annotations.

Remarkably, self-training combined with our verification procedure enables near-perfect detection performance with as few as 6 manually annotated images for LOCO classes and 8 for CELAD. To put this in perspective, the previous state-of-the-art method, PSAD (Kim et al., 2024), relies on 3–5 manually *segmented* samples. Given that bounding box annotations are substantially easier to obtain than pixel-level segmentations, we argue that the supervision required by our framework is comparable, if not lower.

### 4.3 Feature Enrichment

**Distance-based Attention.** While the contextualized object queries produced by the final self-attention layer of the DETR decoder already encode semantic, positional and compositional information, they do not explicitly account for the relative spatial proximity between objects. Intuitively, nearby objects should exert stronger influence when defining each object's context. Since the DETR architecture and training objective impose no such spatial bias, we propose to reinforce local context through a *distance-based attention* mechanism.

Formally, let an image yield $n_d$ detected objects with bounding-box centers $\{c_i \in \mathbb{R}^2\}_{i=1}^{n_d}$ and contextualized embeddings $\{f_i \in \mathbb{R}^d\}_{i=1}^{n_d}$. We compute pairwise distances: $\Delta_{ij} = \|c_i - c_j\|_2$, $\quad \Delta_{ii} = +\infty$, where we manually suppress self-attention. Normalized inverse-distance weights are then defined as:

$$P_{ij} = \frac{(1 + \Delta_{ij})^{-1}}{\sum_{k=1}^{n_d}(1 + \Delta_{ik})^{-1}}.$$

Restricting to the top-$m$ nearest neighbors $\mathcal{N}_i$ of object $i$, we re-normalize:

$$\widetilde{P}_{ij} = \frac{P_{ij}\,\mathbf{1}_{j \in \mathcal{N}_i}}{\sum_{k \in \mathcal{N}_i} P_{ik}}.$$

The context vector of object $i$ is then:

$$g_i = \sum_{j=1}^{n_d} \widetilde{P}_{ij} f_j, \tag{2}$$

**Object Area.** Object size is a critical feature in LAD datasets, yet it is often underrepresented in the embeddings produced by vision backbones. Following prior work, we explicitly enrich each object representation with its estimated area via segmentation. Unlike previous approaches that rely on custom-trained segmentation networks, we leverage the Segment Anything Model (SAM) (Kirillov et al., 2023) in a zero-shot setting. Specifically, the bounding box coordinates predicted by the detector are used as prompts for SAM, which produces binary segmentation masks for the objects. Notably, this approach yields more accurate segmentations than the task-specific networks employed in prior work. Qualitative examples of these results are included in Appendix B.

For each object $i$, with predicted binary segmentation map $p_i$, we compute its area as: $a_i = \sum_{w,h} p_i(w, h)$. Finally, the representative embedding for each object is obtained by concatenating the contextualized embedding $f_i$, the distance-based context $g_i$, and the area feature $a_i$:

$$z_i = [\,f_i; g_i; a_i\,] \in \mathbb{R}^{2d+1}. \tag{3}$$

## 4.4 BIPARTITE MATCHING

Object queries yield slightly different representative embeddings for the same object depending on its compositional context. To robustly measure similarity between images, we propose to compare sets of enriched object embeddings through bipartite matching. Specifically, we maintain a memory bank of representative embeddings from normal images and compute matching distances between a test image and this bank, which later form the basis of our anomaly score.

**Matching Cost.** To solve the matching problem, we first define the pairwise cost between elements of the source and destination sets. Let two images yield sets of representative embeddings

$$Z^{(1)} = \{z_1^{(1)}, \ldots, z_{k_1}^{(1)}\}, \quad Z^{(2)} = \{z_1^{(2)}, \ldots, z_{k_2}^{(2)}\},$$

where each embedding is decomposed as $z = [x; a]$, $x \in \mathbb{R}^{2d}$, $a \in \mathbb{R}_+$. For objects $i$ and $j$, the feature similarity cost is defined as the normalized Euclidean distance between their feature embeddings:

$$C_{ij}^{\text{feat}} = \frac{\|x_i^{(1)} - x_j^{(2)}\|_2}{2d}. \tag{4}$$

Also, to penalize discrepancies in object size, we define:

$$C_{ij}^{\text{area}} = \left| \frac{a_i^{(1)} - a_j^{(2)}}{a_i^{(1)} + a_j^{(2)}} \right|. \tag{5}$$

The total pairwise cost is then: $C_{ij} = C_{ij}^{\text{feat}} + \lambda C_{ij}^{\text{area}}$, where $\lambda > 0$ is a tunable coefficient.

**Linear Assignment Problem.** Given the pairwise cost matrix $C$, we formulate anomaly scoring as a linear assignment problem, seeking the minimum-cost bipartite matching distance. When $k_1 \neq k_2$, some objects remain unmatched. To address this, we pad $C$ with zero-cost entries to form a square matrix of size $\max(k_1, k_2)$, enabling a balanced assignment. We then apply the Hungarian algorithm (Kuhn, 1955), one of the earliest algorithms for balanced linear assignment, to compute

$$D_{\text{matched}}(Z^{(1)}, Z^{(2)}) = \min_{\pi \in \mathcal{M}(k_1, k_2)} \sum_{(i,j) \in \pi} C_{ij}, \tag{6}$$

where $\mathcal{M}(k_1, k_2)$ denotes the set of valid one-to-one matchings. For the remaining unmatched objects, we assign a penalty equal to their maximum similarity cost relative to the opposite set:

$$D_{\text{unmatched}}(Z^{(1)}, Z^{(2)}) = \sum_{i \in U_1} \max_j C_{ij} + \sum_{j \in U_2} \max_i C_{ij}, \tag{7}$$

where $U_1$ and $U_2$ denote the sets of unmatched objects in $Z^{(1)}$ and $Z^{(2)}$, respectively. The complete matching distance is thus: $D = D_{\text{matched}} + D_{\text{unmatched}}$.

**Nearest Neighbor Search.** Given the distance function $D(\cdot, \cdot)$, we define the anomaly score of a test image $I_t$ based on its similarity to the memory bank of normal images $\{I^{(1)}, \ldots, I^{(n)}\}$. Specifically, we identify the $k$ nearest neighbors of $I_t$ under $D$ and aggregate their distances to obtain the final anomaly score:

$$\mathcal{A}(I_t) = \frac{1}{k} \sum_{m \in \text{NN}_k(I_t)} D\left(Z^{(I_t)}, Z^{(I^{(m)})}\right). \tag{8}$$

Figure 2 illustrates the full pipeline.

## 5 EXPERIMENTS

### 5.1 EXPERIMENTAL SETUP

**Implementation Details.** We adopt RT-DETR-large (Zhao et al., 2024) as the base object detector and SAM2.1-base (Ravi et al., 2024) as the zero-shot segmentor. For finetuning, we initialize RT-DETR with COCO-pretrained (Lin et al., 2014) weights, replace the final classification layer to

Table 1: Image-level AUROC comparison on MVTec LOCO AD and CELAD in full-shot and few-shot settings. **Bold** marks the best, underline the second-best, and $^{\dagger}$ denotes results taken from original papers. All values are averaged over five runs.

| Setting | Dataset | SA-PatchCore | ULSAD | EfficientAD | SINBAD | ComAD | CSAD | PSAD | ROMAD |
|---|---|---|---|---|---|---|---|---|---|
| | breakfast box | 86.13 | 84.16 | 85.46$^{\dagger}$ | 97.70$^{\dagger}$ | 94.70$^{\dagger}$ | 94.40$^{\dagger}$ | **100$^{\dagger}$** | 94.80 |
| | juice bottle | 97.04 | 98.80 | 98.41$^{\dagger}$ | 97.10$^{\dagger}$ | 90.90$^{\dagger}$ | 94.90$^{\dagger}$ | **99.10$^{\dagger}$** | 94.06 |
| | pushpins | 64.56 | 86.38 | 97.74$^{\dagger}$ | 88.90$^{\dagger}$ | 89.00$^{\dagger}$ | 99.50$^{\dagger}$ | **100$^{\dagger}$** | 98.66 |
| Full-shot | screw bag | 57.76 | 67.11 | 56.66$^{\dagger}$ | 81.10$^{\dagger}$ | 79.70$^{\dagger}$ | **99.90$^{\dagger}$** | 99.30$^{\dagger}$ | 99.14 |
| | splicing connectors | 89.98 | 87.19 | **95.52$^{\dagger}$** | 91.50$^{\dagger}$ | 84.40$^{\dagger}$ | 94.80$^{\dagger}$ | 91.90$^{\dagger}$ | 85.74 |
| | twin bracelets | 57.60 | 60.90 | 58.12 | 74.68 | 75.38 | 69.26 | 79.48 | **94.16** |
| | average | 75.51 | 80.76 | 81.98 | 88.50 | 85.68 | 92.13 | **94.96** | 94.43 |
| | LOCO | 73.75 | 72.11 | 68.92 | 84.00 | 81.78 | 69.40 | 85.59 | **92.44** |
| 32-shot | twin bracelets | 55.98 | 58.31 | 53.47 | 71.71 | 65.00 | 54.90 | 61.75 | **85.67** |
| | average | 70.79 | 69.81 | 66.35 | 81.95 | 78.98 | 66.99 | 81.62 | **91.32** |
| | LOCO | 71.61 | 72.19 | 67.87 | 82.68 | 77.22 | 62.69 | 75.25 | **91.96** |
| 16-shot | twin bracelets | 54.13 | 54.15 | 48.41 | 72.34 | 65.30 | 52.55 | 51.80 | **80.24** |
| | average | 68.70 | 69.18 | 64.63 | 80.95 | 75.23 | 61.00 | 71.34 | **90.01** |
| | LOCO | 67.77 | 60.82 | 64.25 | 82.00 | 76.10 | 61.83 | 65.27 | **90.84** |
| 8-shot | twin bracelets | 52.48 | 50.03 | 52.88 | 67.46 | 60.90 | 52.73 | 48.88 | **70.60** |
| | average | 65.22 | 59.02 | 62.35 | 79.58 | 73.57 | 60.31 | 62.54 | **87.47** |

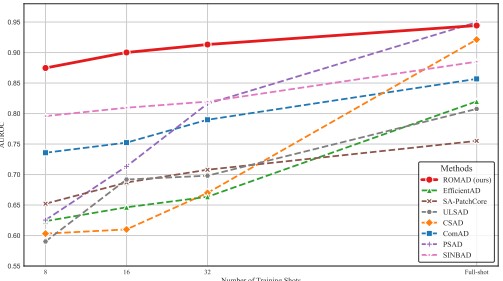

Figure 3: Average AUROC of different methods in full-shot and few-shot settings.

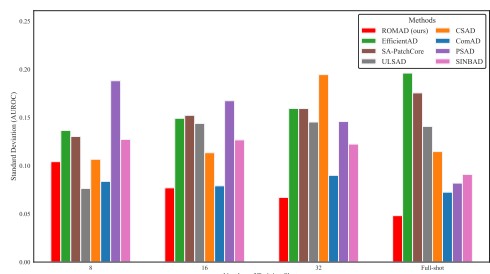

Figure 4: Standard deviation of results across all datasets, illustrating the stability and generalization of each method.

match our dataset classes, and then perform full-network finetuning. We use 6 manually annotated samples for each class of MVTec LOCO AD (Bergmann et al., 2022) and 8 samples for CELAD. Training is performed in two passes via self-training, each for 600 epochs with early stopping. For CELAD, we additionally employ intra-image Copy-Paste augmentation (Ghiasi et al., 2021; Dwibedi et al., 2017; Kisantal et al., 2019) during the first training pass. In the verification procedure, we fix $\tau_{\text{ver}} = 0$, such that only pseudo-labels with exactly matching label histograms to the reference set are retained. After training, we extract the contextualized object queries from the outputs of the 6th self-attention layer of the DETR decoder. In the distance-based attention module, the maximum number of neighbors is set to $m = 3$ for all datasets. The area coefficient is fixed at $\lambda = 0.5$.

Although implemented on RT-DETR, in principle, our framework can be applied to any attention-based detector that produces self-attended object embeddings, such as YOLOv10 (Wang et al., 2024) or Sparse R-CNN (Sun et al., 2021), highlighting the versatility of our approach.

**Datasets and Baselines.** We evaluate our approach on CELAD and the logical subset of LOCO, covering six logically distinct datasets. Anomaly detection performance is measured using image-level AUROC. For our method, we additionally report AP and F1-max in Appendix B. We compare against seven baselines: PSAD (Kim et al., 2024), CSAD (Hsieh & Lai, 2024), ComAD (Liu et al., 2023), SINBAD (Cohen et al., 2023), EfficientAD (Batzner et al., 2024), ULSAD (Patra & Taieb, 2024), and SA-PatchCore (Ishida et al., 2023). Among these, the two strongest baselines incorporate additional supervisions. PSAD relies on 3–5 manually segmented samples and CSAD employs text supervision for open-vocabulary object segmentation. In total, we conducted **715 baseline runs**, which to the best of our knowledge cover all LAD baselines with publicly available implementations.

Table 2: Component ablation.

| Area | Distance-based attention | NMS | Unmatched policy | LOCO | CELAD | Average |
|------|--------------------------|-----|------------------|------|-------|---------|
| ✗ | ✗ | ✗ | max | 92.15 | 85.25 | 91.00 |
| ✓ | ✗ | ✗ | max | 94.32 | 89.56 | 93.53 |
| ✓ | ✓ | ✗ | max | 94.07 | 94.08 | 94.07 |
| ✓ | ✓ | ✓ | max | **94.48** | **94.16** | **94.43** |
| ✓ | ✓ | ✓ | min | 88.80 | 92.98 | 89.50 |

Table 3: Effect of self-training.

| Dataset | Pass 1 # verified samples | Pass 2 # verified samples |
|---------|---------------------------|---------------------------|
| breakfast box | 347/351 | 350/351 |
| juice bottle | 334/335 | 335/335 |
| pushpins | 370/372 | 372/372 |
| screw bag | 347/360 | 359/360 |
| splicing connectors | 359/360 | 360/360 |
| twin bracelets | 223/340 | 339/340 |

## 5.2 MAIN RESULTS

Table 1 compares ROMAD with prior baselines under both full-shot and few-shot settings. Across LOCO and CELAD, ROMAD ranks second overall, with only a 0.5% gap from the top-performing method. On CELAD specifically, it achieves a new state of the art, outperforming the next-best method by 14.7%.

**Few-shot Comparison.** Evaluating LAD methods in few-shot regimes is crucial for two reasons: (i) most practical scenarios provide only a limited number of training samples, and (ii) few-shot performance serves as a strong indicator of genuine logical understanding, as is standard in evaluation protocols across other logical domains. In such cases, systems are expected to capture the underlying rules rather than memorize all possible instances. In this regime, ROMAD achieves the best average performance across both LOCO and CELAD, with a clear margin of up to 9.4% over the strongest baseline. Figure 3 summarizes the average results for full-shot and few-shot settings, highlighting the consistent gap between ROMAD and competing methods. Figure 4 reports the standard deviation across six datasets, reflecting each method's stability. Notably, ROMAD achieves substantially lower variance, indicating a more balanced generalization across datasets.

## 5.3 ABLATION STUDY

We validate the effectiveness of the different components of ROMAD. Table 2 reports the impact of feature enrichment with object area and distance-based attention, as well as applying NMS in object detection, showing performance gains with each added module. Most of the gains from the distance-based attention module are observed on CELAD, since most LOCO classes do not require a strict spatial arrangement of elements. We also experimented with different policies for the anomaly score associated with unmatched elements and found that using the maximum similarity cost yielded the best results.

Table 3 evaluates the effect of self-training in achieving strong detection performance with only a few manually annotated samples. After the second self-training pass, we reach almost perfect detection performance in terms of the verification score, highlighting both the power of existing object detectors in modeling complex scenes and the effectiveness of self-training in further reducing annotation requirements. It is also worth noting that the number of manually annotated samples for LOCO was not heavily optimized and could potentially be reduced even further.

## 6 CONCLUSION

In this work, we addressed the current limitations of existing LAD research by introducing CELAD, a new benchmark explicitly designed to test compositional understanding. CELAD substantially increases the complexity of LAD by featuring more objects per scene, greater variability, and more intricate anomaly types. Our evaluation demonstrates that state-of-the-art methods, which achieve near-saturated results on previous benchmarks, experience notable drops on CELAD, underscoring the need for more generalizable approaches. We also proposed ROMAD, a relation-aware object matching framework that leverages DETR's relation modeling capabilities alongside a lightweight matching pipeline. Despite its simplicity, ROMAD achieves a new state of the art on CELAD with a 14.7% margin, maintains competitive results on prior datasets, and delivers the strongest performance in few-shot regimes. We view ROMAD as a minimal baseline for LAD, intended to support our new benchmark and the argument that current models lack in generalization capabilities.

REPRODUCIBILITY STATEMENT

We release the full implementation of our method as well as the CELAD dataset for review. Since we report results for seven baselines across datasets and settings that were not necessarily covered in the original papers, we also provide the modified code used to reproduce these baselines. The modifications to the official implementations are minimal and limited to adding support for CELAD data loaders and few-shot settings. Code and data are available for review at: `https://github.com/neutral-coder-737/Home-Page`. For confidentiality, the dataset is provided as an encrypted file, with the password included in the supplementary materials for reviewers. The dataset will be made publicly available upon acceptance.

ETHICS STATEMENT

We used large language models (LLMs) solely for language editing purposes, such as rephrasing and polishing the final text of the paper. No parts of the technical content, experiments, or results were generated by LLMs.

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

# A   DATASET

## A.1   DATASET DESCRIPTION

CELAD consists of images of two bracelet types. Both types feature letter beads spelling "SPARK," but with distinct color combinations. In Twin A, the sequence begins with a black bead carrying a white "S," followed by alternating color patterns for the remaining letters. Twin B follows the exact opposite color pattern for its letter beads. The letter beads are separated by small hexagonal patterned beads, and each bracelet type has a slightly different arrangement of black and white beads. Variations produced by rotations, flips, and bendings of the bracelets are all considered normal, giving rise to substantially greater diversity among the normal samples.

## A.2   DATASET STATISTICS

Different approaches to LAD have varying strengths and weaknesses in detecting specific types of anomalies. It is therefore crucial for benchmarks to encompass diverse anomaly types so as not to favor a particular family of methods. CELAD includes a considerably broad range of anomaly types—covering semantic, metrological, and relational features—thereby providing a more comprehensive assessment of a model's logical understanding. Table 4 presents detailed statistics of CELAD, including the number of samples and sub-types within each anomalous category. In particular, anomaly types such as wrong orientations or wrong permutations are especially challenging, since they cannot be resolved through a simple matching of object categories.

Table 4: Overview of CELAD dataset statistics, including five distinct anomaly types, each with multiple sub-types.

|  | Normal | Cross Combinations | Extra Beads | Wrong Orientations | Duplicate Letters | Wrong Permutations |
|---|---|---|---|---|---|---|
| # samples | 530 | 20 | 34 | 71 | 66 | 29 |
| # sub-types | 2 | 2 | 5 | 10 | 10 | 4 |

## A.3   PIXEL-LEVEL GROUND TRUTHS

Pixel-level anomaly scores have long served as the primary means of evaluating explainability in structural AD, where most state-of-the-art methods already achieve strong performance on pixel-level benchmarks (Bergmann et al., 2019; Zou et al., 2022). In LAD, the pixel-level evaluation protocol largely follows the same conventions as structural AD, with the added complexity that multiple valid ground-truth annotations may exist for a single anomalous image (Bergmann et al., 2022). Figure 5 illustrates this by showing examples of anomalous samples in CELAD together with all of their valid ground-truth annotations.

## A.4   LIMITATIONS OF PIXEL-LEVEL EVALUATION FOR LAD

Although some existing LAD methods demonstrate near-perfect performance at the image level, explainability remains an underexplored direction. Unlike in structural AD, where pixel-level anomaly maps are both common and effective, many high-performing LAD methods—including ours—do not produce pixel-level outputs. While anomaly heatmaps provide a useful form of interpretability for structural anomalies, they are particularly well-suited to settings where anomalies are localized

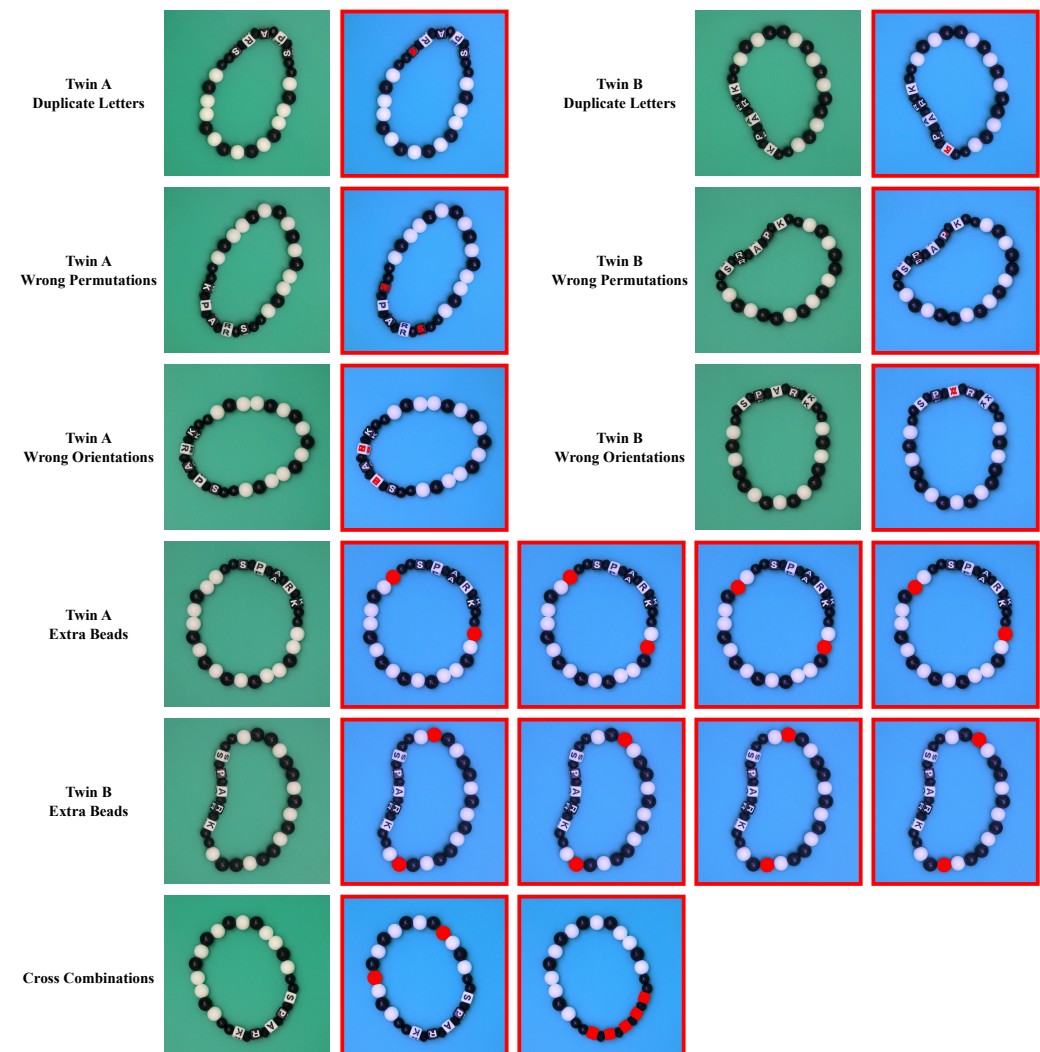

Figure 5: Examples of CELAD anomaly types along with their pixel-precise ground-truth annotations. Each row shows anomalous images alongside all its valid annotations, whose number varies by instance.

to specific regions of an image. This assumption does not necessarily hold for logical anomalies, where defects are often distributed across objects or relations and may admit multiple valid answers. We therefore argue that conventional pixel-level evaluation is not the ideal measure of explainability for LAD, and we leave it to future work to develop alternative evaluation protocols tailored to this setting.

## A.5 DISTINCTION BETWEEN STRUCTURAL AND LOGICAL ANOMALIES

Most prior works on LAD report results jointly over both structural and logical anomalies. However, in practice, these two types of anomalies are often handled through separate branches or mechanisms (Bergmann et al., 2022; Batzner et al., 2024; Sugawara & Imamura, 2024; Hsieh & Lai, 2024; Zhang et al., 2024a), suggesting that the underlying challenge of structural and logical AD differ substantially. Furthermore, the nature of logical anomalies in real-world scenarios is far more complex than what existing benchmarks cover. By introducing CELAD, we highlight this gap and demonstrate that current LAD methods struggle to understand and reason about more complex scenes.

Taken together, these points suggest that logical AD is inherently different and more difficult than structural AD and merits its own dedicated line of research.

# B  METHOD

## B.1  IMPLEMENTATION DETAILS

We use RT-DETR-large (Zhao et al., 2024) as our base object detector and SAM2.1-base (Ravi et al., 2024) as the zero-shot segmentor, both through the implementations provided by Jocher et al. (2023). RT-DETR-large has ~33M parameters, and SAM2.1-base has ~81M parameters, reflecting the minimal memory consumption of ROMAD.

For fine-tuning the object detector, we follow the established convention of initializing from COCO-pretrained (Lin et al., 2014) weights, replacing the last classification layer to match the number of classes in the target dataset, and then performing full fine-tuning. Training proceeds in two passes via self-training, with the validation set identical to the training set. Each pass runs for up to 600 epochs with early stopping. For CELAD, we additionally apply intra-image Copy-Paste augmentation (Ghiasi et al., 2021; Dwibedi et al., 2017; Kisantal et al., 2019), but only during the first training pass to support the detector's classification objective for pseudo-label generation.

In the verification procedure, we fix $\tau_{\text{ver}} = 0$, retaining only pseudo-labels with exactly matching label histograms. After self-training, we extract contextualized object queries from the outputs of the 6th self-attention layer of the DETR decoder. We use the same hyperparameters across all datasets: the confidence threshold for object detection is set to $\tau_{\text{conf}} = 0.7$, the maximum number of neighbors for the distance-based attention module is $m = 3$, the area coefficient is fixed at $\lambda = 0.5$, and the number of nearest neighbors for the final KNN search is $k = 3$.

Further details for each step are discussed in the following sections. All experiments on our method and other baselines are conducted on a single NVIDIA RTX 3090 24GB GPU.

## B.2  CHOICE OF OBJECT DETECTORS

The main distinction of the DETR family of object detectors (Carion et al., 2020; Zhu et al., 2020; Meng et al., 2021; Zhao et al., 2024) compared to conventional detectors lies in their transformer-based architecture and the use of self-attention (Vaswani et al., 2017) among object queries. While DETR was the first to introduce this design, subsequent object detectors have increasingly incorporated variants of self-attention. For instance, Sparse R-CNN (Sun et al., 2021) applies self-attention to its set of object features prior to the dynamic instance interaction heads, enabling reasoning over inter-object relations. Similarly, YOLOv10 (Wang et al., 2024) adopts a Partial Self-Attention (PSA) module, which uses the global modeling capabilities of self-attention while reducing its quadratic computational cost. Given this trend, we believe our lightweight matching pipeline could be readily extended to any object detector that produces self-attended object embeddings.

## B.3  VERIFICATION THRESHOLD

The verification threshold $\tau_{\text{ver}}$ is selected using a dynamic quantile-based strategy. However, as shown in Table 3, even in the most challenging case (i.e., CELAD), 66% of the predicted annotations from the first pass satisfy $d_{\text{ver}}(h^{(p)}) = 0$. Given the satisfactory performance of the first pass, coupled with the fact that in our extremely low-data regime the quality of each pseudo-label is especially important, we fix the verification threshold at $\tau_{\text{ver}} = 0$.

## B.4  DATA AUGMENTATION

CELAD introduces substantially more object categories than MVTec LOCO AD (Bergmann et al., 2022), with subtle inter-class differences. Additionally, CELAD contains two bracelet types, meaning that some objects of the training set appear as few as four times. To address this rarity, we adopt an augmentation strategy focused on underrepresented objects of the scene. We employ intra-image Copy-Paste augmentation in CELAD to increase the frequency of rare objects. Concretely, from the set of manually annotated samples with reference histograms $\{h_1^{(r)}, \ldots, h_K^{(r)}\}$, we identify rare cat-

egories as those with the lowest values of $\sum_{k=1}^{K} h_{k,c}^{(r)}$. For each reference image, we then randomly select two or three such objects, copy their instances (using the annotated bounding boxes), and paste them randomly within the same image. Remarkably, this simple strategy raised the number of verified samples of the first pass from 20/340 to 223/340.

It is important to mention that this augmentation is applied only in the first pass to support the detector's classification objective during pseudo-label generation. We avoid using it in the second pass, since random placement of objects could disrupt the model's ability to capture the true compositional structure of the scene.

## B.5 ADDITIONAL EVALUATION METRICS

AUROC is the most widely used metric for evaluating AD performance, but it cannot represent the true performance of models in all situations. For instance, in imbalanced datasets where the ratio of normal to anomalous samples is skewed, AUROC alone may present an inflated view of performance (Davis & Goadrich, 2006; Saito & Rehmsmeier, 2015; Cook & Ramadas, 2020). Table 5 reports all three AD metrics for ROMAD—AUROC, AP, and F1-max—across all datasets.

Table 5: Complete quantitative results of ROMAD.

| Setting | Metric | breakfast box | juice bottle | pushpins | screw bag | splicing connectors | twin bracelets | average |
|---|---|---|---|---|---|---|---|---|
| Full-shot | AUROC | 94.80 | 94.06 | 98.66 | 99.14 | 85.74 | 94.16 | 94.43 |
| | AP | 95.42 | 96.96 | 98.48 | 99.09 | 88.01 | 95.68 | 95.61 |
| | F1-max | 88.46 | 91.10 | 94.92 | 96.82 | 79.61 | 88.89 | 89.97 |
| 32-shot | AUROC | 91.73 | 91.39 | 98.13 | 99.02 | 81.96 | 85.67 | 91.32 |
| | AP | 93.05 | 95.10 | 97.65 | 98.92 | 86.23 | 87.18 | 93.02 |
| | F1-max | 87.57 | 88.42 | 92.55 | 97.47 | 76.00 | 81.32 | 87.22 |
| 16-shot | AUROC | 89.17 | 92.06 | 98.07 | 98.36 | 82.16 | 80.24 | 90.01 |
| | AP | 90.60 | 95.06 | 97.82 | 98.22 | 85.73 | 81.58 | 91.50 |
| | F1-max | 80.82 | 88.57 | 93.26 | 96.38 | 76.19 | 78.52 | 85.62 |
| 8-shot | AUROC | 91.76 | 84.90 | 98.28 | 96.94 | 82.32 | 70.60 | 87.47 |
| | AP | 92.42 | 91.55 | 98.17 | 94.59 | 85.58 | 75.79 | 89.68 |
| | F1-max | 83.22 | 81.53 | 94.19 | 96.06 | 73.68 | 72.60 | 83.55 |

## B.6 OBJECT DETECTION AND SEGMENTATION RESULTS

Figure 6 presents the object detection and segmentation outputs of ROMAD. Notably, our method produces more accurate annotations than the task-specific networks employed in prior work. For example, CELAD contains visually similar object classes that are difficult to distinguish. Although these cases pose challenges to baseline methods, ROMAD achieves near-perfect detection performance on them.

## B.7 MODEL ADJUSTABILITY

As with other segmentation-based methods, the way images are annotated has a direct impact on the final performance. We argue that methods relying on a small number of manually annotated samples, such as ours, offer additional adjustability to the end user. Specifically, they allow users to define which objects in a scene are most relevant for capturing the underlying logical rules, providing a means to adjust the model to meet customized requirements. This property is especially important in practice, since the definition of human-perceived anomalies is inherently subjective and may not always align perfectly with statistical deviations from the normal distribution. For example, while the model might flag inconsistencies in the background as anomalous, humans may consider them irrelevant to the primary objects of interest (Liu et al., 2023).

## B.8 SIMPLICITY

Our method is among the simplest approaches for LAD, requiring no custom architectures or heavily optimized objectives. Instead, we leverage existing vision foundation models in few-shot and

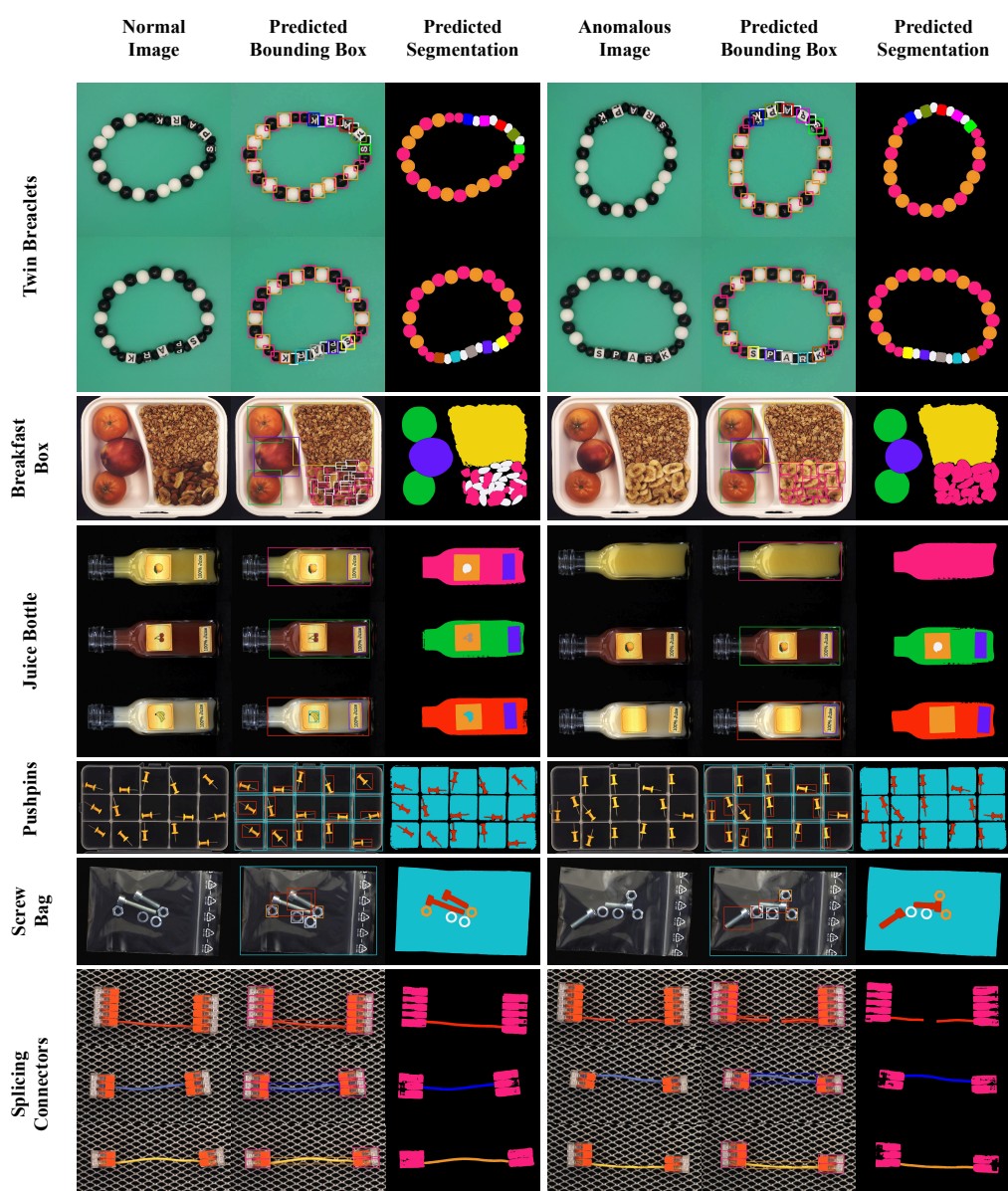

Figure 6: Qualitative results of ROMAD's object detection and segmentation. Each row shows the annotations for both normal and anomalous samples.

zero-shot settings, combined with an intuitive object-matching pipeline. Despite this simplicity, our method achieves competitive performance and more balanced results across diverse datasets, indicating stronger generalization capabilities.

We view our approach as a basic and minimal baseline for LAD, intended to support our new benchmark and the argument that current models are overfitted to existing datasets, highlighting the need for more generalizable approaches. We believe that future methods—equipped with specially optimized architectures and training objectives—have the potential of achieving significantly higher performance.

### B.9 LIMITATIONS

ROMAD does not address structural anomalies, which is a limitation of our method. However, as discussed earlier, we believe that LAD warrants dedicated methods, and in practice, the joint detection of structural and logical anomalies could be achieved by ensembling specialized methods for each. Furthermore, although the matching-based design of ROMAD allows for explainable anomaly detection, it does not produce pixel-level anomaly maps.

In addition, Table 6 reports the misclassifications of ROMAD on CELAD when using the optimal threshold (i.e., the one yielding the highest F1 score) to discretize image-level anomaly scores. While ROMAD successfully detects all instances across CELAD anomaly categories, it shows reduced performance in detecting the particularly challenging case of anomalies caused by wrong orientations.

Table 6: Misclassifications of ROMAD on CELAD.

|  | Normal | Cross Combinations | Extra Beads | Wrong Orientations | Duplicate Letters | Wrong Permutations |
|---|---|---|---|---|---|---|
| # test samples | 190 | 20 | 34 | 71 | 66 | 29 |
| # misclassified samples | 25 | 0 | 0 | 24 | 0 | 0 |

## C BASELINES

### C.1 CHOICE OF BASELINES

Since we evaluate prior methods on a new dataset as well as in few-shot settings, many results are not directly available from the original papers. We therefore restrict our comparison to baselines with publicly released implementations. To the best of our knowledge, baselines included in Table 1 cover all existing LAD methods with accessible codebases (Kim et al., 2024; Hsieh & Lai, 2024; Liu et al., 2023; Cohen et al., 2023; Batzner et al., 2024; Patra & Taieb, 2024; Ishida et al., 2023).

We exclude LLM-based approaches (Zhang et al., 2025; 2024c) from our comparison, as their settings are fundamentally different: they rely on large foundation models and substantially greater computational resources, making them impractical for real-time applications where efficiency is critical.

### C.2 EVALUATION SETUP

In total, considering both few- and full-shot settings, multiple datasets, and five independent runs per result, we conducted a substantial number of **715 baseline runs**, representing a large-scale experimental effort to ensure the reliability of our comparisons. We are the first to evaluate such a broad set of models across so many settings. Our results reveal a clear reordering of methods in few-shot scenarios: state-of-the-art approaches that dominate in the full-shot setting often perform worse in low-data regimes. This indicates that simpler and more general models such as SINBAD (Cohen et al., 2023) and ComAD (Liu et al., 2023) capture the underlying relations more effectively, while full-shot state-of-the-art methods tend to rely heavily on memorization of training instances.

### C.3 REPRODUCTION DETAILS

Among the baselines, PSAD (Kim et al., 2024) and CSAD (Hsieh & Lai, 2024) incorporate additional supervision. Below, we detail the manual supervision provided for each:

- PSAD relies on a small number of manually segmented samples to train its segmentation network. According to the original paper, for MVTec LOCO AD (Bergmann et al., 2022) classes with multiple types (e.g., three types within "juice bottle" and "splicing connectors"), the authors use one manually segmented sample per type, totaling three. For CELAD, which has two bracelet types, we use two manually segmented samples per type for fair comparison, totaling four. Moreover, the number of components is set to 15 to match CELAD's 15 distinct bead categories.

- CSAD relies on text prompts for open-vocabulary object segmentation (Liu et al., 2024). For CELAD, the prompts provided are: "black bead", "white bead", "hexagon", "s", "p", "a", "r", "k", following the authors' convention for LOCO.

