# OpenReview forum: "CELAD: Compositional Evaluation for Logical Anomaly Detection"
_ICLR.cc/2026/Conference — Submitted to ICLR 2026_

### Official Review · Reviewer_jtKz · 2025-10-30

**Soundness:** 2
**Presentation:** 3
**Contribution:** 2
**Rating:** 2
**Confidence:** 4

**Summary:**

Paper contributes a new dataset, CELAD, for the logical anomaly detection (LAD) problem, which contains images of beaded bracelets. Each bracelet is defined by the letter beads spelling "SPARK" and a combination of black and white colored beads. There are 530 normal samples and 220 anomalous samples across 5 distinct anomaly categories.

It also proposes a novel LAD method based on a DETR object detector - ROMAD. ROMAD uses DETR to detect the objects of interests (OOI) and compare the detected OOI with a set of pretrained memory bank for anomaly detection. The logical pattern of the numerical distributions of normal OOIs are represented as a reference histogram and L2 distance is computed for the inferred histogram and the reference histograms to detect anomalous arrangement of OOIs.

Each reference image is also represented as set of representative embeddings of the OOIs, the set of embeddings are then linearly assigned to the detected embeddings. The complete Distance is thus: D_matched + D_unmatched.

**Strengths:**

Presentation: Paper is well written and easy to follow.

Motivations: Motivations are clearly explained and supported by the proposed model and the experimental setup.

Methodology: Methodology is sound and efficient.

**Weaknesses:**

1. Dataset is only limited to beaded bracelets and does not represent the complexity of the real world problem of LAD.

2. The proposed method is specifically designed for the specific type of LAD. There is no formal proof, nor experimental results to support that the proposed method is generalizable across different types of LAD.

3. Experimental results show that SOTA methods outperform the proposed method in the full-shot setting.

4. Novelty and contributions are limited.
- The dataset is only for one specific type of objects.
- Methodology performance

**Questions:**

1. Dataset is not sufficiently representative of the LAD diversity and complexity.
2. Proposed method does not outperform SOTA in the full-shot setting.

Please comment on the above.

---

> ### Author Response · Authors · 2025-11-21
>
> Dear Reviewer jtKz,
>
> Thank you for your constructive comments on our paper. Below, we provide a detailed response to your questions and comments. Please let us know if any concerns remain insufficiently addressed; we would be happy to further clarify.
>
>
> > **W1 & Q1: Dataset scope and representativeness**
>
> We consider your concern as addressing two separate issues.
>
>  1. **The dataset is limited to a single object category (beaded bracelets).**
>
>     We have explicitly stated that CELAD is an additional benchmark intended to complement the already existing datasets. We do not claim it to be a self-sufficient benchmark and we strongly encourage future works to evaluate across all available datasets to demonstrate maximum generalizability. In line with this principle, we have evaluated our method not only on CELAD but also on MVTec LOCO.
>
>     Nevertheless, we want to emphasise the fact that in LAD, simply adding more object categories does not necessarily increase task difficulty. In contrast to structural AD—where anomaly types share a common logic—LAD requires each category to present distinct logical constraints. Therefore, the diversity of anomaly types is more important than the diversity of object categories. CELAD introduces 31 logically distinct anomaly types—on average five times more than a LOCO class—representing a substantial expansion of logical complexity not previously covered in any LAD dataset.
>
>  2. **The dataset not representing real-world LAD complexity.**
>
>     Frankly, advancing the complexity of LAD benchmarks is the main motivation of our work. We show that SOTA LAD methods, which achieve near-saturated performance on prior benchmarks, suffer drops of over 20% on CELAD. Below, we include a table comparing the performance of prior LAD baselines on MVTec LOCO and CELAD to more clearly illustrate the performance gap introduced by these datasets.  As the table shows, CELAD introduces substantially greater evaluation complexity.
>
>     While we agree that additional LAD benchmarks are still needed, within the context of existing datasets, CELAD introduces a clear step forward in difficulty and realism.
>
>     | Dataset Class              | Average AUROC Across Previous Baselines | Maximum AUROC Across Previous Baselines |
>     |---------------------------|-------------------------------------------|-------------------------------------------|
>     | breakfast box             | 91.79                                     | 100                                       |
>     | juice bottle              | 96.61                                     | 99.1                                      |
>     | pushpins                  | 89.44                                     | 100                                       |
>     | screw bag                 | 77.36                                     | 99.9                                      |
>     | splicing connectors       | 90.76                                     | 95.52                                     |
>     | twin bracelets (CELAD)    | 67.92                                     | 79.48                                     |
>
> &nbsp;
>
> > **W2: Generalizability of the proposed method across different types of LAD**
>
> As stated in our experiments section, our method is evaluated on both CELAD and MVTec LOCO, covering six logically distinct datasets. Compared to prior works that report results solely on MVTec LOCO, our conducted evaluations are the most comprehensive evaluation of LAD methods to date.
> Our results demonstrate that ROMAD generalizes across different LAD scenarios and is not tailored specifically to our dataset but is applicable across multiple forms of logical anomalies.

---

> ### Author Response · Authors · 2025-11-21
>
> > **W3 & Q2: Performance relative to SOTA in the full-shot setting**
>
> While being competitive, ROMAD is not claimed to have a full-shot SOTA performance. As stated throughout the paper, our main contribution with ROMAD is to introduce a simple yet effective baseline that is **significantly more generalizable than the current methods**. Our primary aim is to highlight generalization gaps in existing LAD methods that perform strongly on LOCO yet degrade significantly—by over 20%—on CELAD. In contrast, ROMAD achieves highly consistent performance across datasets (94.48% AUROC on LOCO and 94.16% on CELAD), exhibiting the lowest performance variance among all baselines (Fig. 4). This stability is a key advantage in LAD, where robustness across settings is essential.
>
> Moreover, ROMAD achieves SOTA performance in the few-shot regime by a substantial margin. Given that few-shot AD has an established research community and is viewed as a meaningful standalone contribution, we emphasise not to overlook the few-shot contribution of our method. To our knowledge, no existing LAD method achieves decent few-shot performance, let alone also being competitive in the full-shot regime.
>
> Our work therefore serves as a reevaluation of the field by showing that a simple, training-free pipeline can:
>  - achieve SOTA few-shot performance,
>  - maintain competitive full-shot performance, and
>  - exhibit substantially stronger generalization than heavily customized architectures.
>
> &nbsp;
>
> > **W4: Novelty and overall contributions**
>
> In previous sections, we aimed to clarify your concerns about our contributions regarding both our dataset and our methodology. Here we summarize our contributions and their impact as follows:
>  - **A novel real-world LAD dataset** introducing a wide range of novel anomalies not covered before.
>  - **A new method leveraging relational modeling of DETR**, a perspective not explored in prior work on LAD.
>  - **Improved generalization**, evidenced by consistent performance across existing and newly introduced benchmarks.
>  - **Highest performance on CELAD** with a margin of 14.7%.
>  - **SOTA few-shot results** with improvements of up to 9.4%.
>  - **An extensive experimental study** involving a substantial number of 715 baseline runs—the most comprehensive evaluation in the LAD literature to date.
>
> While much prior work in LAD focuses on incremental model improvements over the existing datasets, our contribution is broader: we introduce a large-scale reevaluation that highlights fundamental gaps and future directions in a field that is already saturated, yet possesses great potential.

---

### Official Review · Reviewer_4exq · 2025-10-31

**Soundness:** 3
**Presentation:** 2
**Contribution:** 3
**Rating:** 4
**Confidence:** 4

**Summary:**

This paper introduces CELAD, a new benchmark for logical anomaly detection (LAD) that emphasizes compositional and relational rule violations rather than simple structural defects. It also proposes ROMAD, a DETR-based framework that performs training-free anomaly matching with minimal annotations. Experiments show that ROMAD achieves state-of-the-art results on CELAD

**Strengths:**

1.CELAD is a more challenging LAD benchmark with richer compositional rules and diverse anomalies, exposing poor generalization of current SOTA methods.
2.ROMAD is simple, few-shot efficient, and leverages DETR’s relational embeddings with a training-free matching pipeline.

**Weaknesses:**

1.Ignores structural anomalies, limiting real-world applicability where both anomaly types coexist.

2.No pixel-level anomaly maps, reducing interpretability and compatibility with standard AD evaluation.

3.Annotation cost comparison with segmentation-based methods lacks empirical justification.

**Questions:**

1.Why does ROMAD underperform slightly on LOCO despite strong results on CELAD? Is LOCO too simple to benefit from relational modeling?

2.How sensitive is the distance-based attention to object density variations across datasets?

3.Are there plans to extend CELAD to more object categories or dynamic (video) settings?

If my main concerns are properly addressed, I would be willing to raise my evaluation.

---

> ### Author Response · Authors · 2025-11-21
>
> Dear Reviewer 4exq,
>
> Thank you for your thoughtful and valuable comments. We address each of your concerns in detail below. If any point remains unclear, we would be happy to follow up.
>
> > **W1: Why we did not focus on structural AD**
>
> Regarding the absence of structural AD results, our paper consistently motivates the importance of treating LAD as a distinct research direction. Our decision is based on several observations:
>
>  1. **Prior works already separate structural and logical AD into two independent components.**
>
>     Most existing methods—including PSAD, CSAD, GCAD, EfficientAD, PUAD, DSKD, and ULSAD—explicitly handle structural and logical anomalies through two separate branches. These branches are either completely independent or share only minimal components such as the image encoder. To highlight this more concretely:
>
>     - **PSAD** constructs its anomaly score from three memory banks. The “patch representation memory bank,” which is responsible for detecting fine-grained structural anomalies, is directly formed by using pixel-level scores of PatchCore, a well-established structural AD method.
>
>     - **CSAD** incorporates a dedicated “patch histogram branch” for logical anomalies and a separate “LGST branch” responsible for fine-grained structural anomalies. As stated in their paper, the LGST branch is a distillation-based structural AD module adapted from EfficientAD with only minor efficiency-related modifications.
>
>     These examples illustrate that current “unified” methods are typically rebranded ensembles of a logical module and a structural module, rather than genuinely integrated architectures. A truly unified model capable of jointly solving both tasks with competitive performance remains non-existent in the literature.
>
>  2. **LAD is the weaker link.**
>
>     Our dataset demonstrates that the logical anomaly types encountered in real-world scenarios are far more challenging than those previously covered, showcasing that current LAD approaches lag significantly behind structural AD methods. We therefore believe that focusing research effort on the more difficult and underexplored component—LAD—is both timely and impactful.
>
>     We also note that ComAD, an efficient and well-established method, is LAD-only, further indicating that LAD-focused methods are neither uncommon nor limited in applicability.
>
>  3. **Structural and logical anomalies require fundamentally different inductive biases.**
>
>     The nature of structural anomalies differs drastically from the relational understanding required for logical anomalies. This fundamental distinction explains why most current methods treat the two tasks as mixtures of separate models. It also suggests that attempting to unify them without addressing LAD-specific challenges first may be premature. Given that LAD is considerably more challenging than structural AD, a dedicated LAD method is no more restrictive in scope than existing structural-only approaches. Our work therefore concentrates on addressing what we view as the current bottleneck in AD research.
>
> Nonetheless, we verify compatibility with a structural branch. Following the aforementioned convention, we provide the results from ensembling our method with a structural branch (using RD4AD) on the MVTec AD dataset:
>
> | Dataset Class | Structural Branch | Logical Branch | Combined |
> |---------------|-------------------|----------------|----------|
> | Grid         | 99.7 | 78.5 | **99.7** |
> | Cable        | 96   | 68.4 | **96.2** |
> | Capsule      | 97.6 | 58.1 | **97.6** |
> | Bottle       | 99.9 | 67.7 | **100** |
> | Tile         | 99.9 | 88.2 | **99.9** |
> | Screw        | 98.7 | 54.2 | **98.5** |
> | Wood         | 99.4 | 87.7 | **99.5** |
> | Leather      | 100  | 84.2 | **100** |
> | Metal Nut    | 100  | 72.1 | **100** |
> | Zipper       | 98.6 | 91.1 | **99** |
> | Hazelnut     | 100  | 51.2 | **100** |
> | Transistor   | 96.9 | 92.6 | **99.8** |
> | Toothbrush   | 97.5 | 66.7 | **97.5** |
> | Pill         | 96.8 | 50.4 | **96.8** |
> | Carpet       | 98.6 | 91.9 | **99.2** |
>
> Please Note that the choice of RD4AD for our structural branch is not optimized for maximal performance due to the time constraints during the rebuttal phase. The results are just to demonstrate that our logical branch can be seamlessly combined with a structural branch, and the outputs of branches do not degrade one another.

---

> ### Author Response · Authors · 2025-11-21
>
> > **W2:  Lack of pixel-level anomaly maps**
>
> Unlike conventional structural AD where generating pixel-level anomaly maps has become a standard, in LAD, the field has not yet converged on providing meaningful forms of pixel-level interpretability. Among our list of baselines, only ULSAD and EfficientAD report pixel-level evaluations, whereas other methods—including PSAD, CSAD, ComAD, SINBAD, and SA-PatchCore—do not produce pixel-level logical anomaly maps. **This reflects a broader limitation of the field rather than a limitation of ROMAD specifically.**
>
> We also have a dedicated discussion on the limitations of pixel-level evaluation for LAD in Appendix A. While anomaly heatmaps provide a useful form of interpretability for structural anomalies, they are particularly well-suited to settings where anomalies are localized to specific regions of an image. This assumption does not necessarily hold for logical anomalies, where defects are often distributed across objects or relations and may admit multiple valid answers.
> We therefore argue that conventional pixel-level evaluation is not necessarily the ideal measure of explainability for LAD, and we motivate the need to develop alternative evaluation protocols tailored to LAD.
>
> &nbsp;
>
> > **W3: Annotation cost comparison**
>
> Our statement regarding annotation cost stems from our direct empirical experience: for every sample where we annotated bounding boxes, producing pixel-precise segmentation masks required at least three times more annotation time. It is widely accepted that segmentation supervision imposes substantially higher annotation cost compared to four point bounding boxes.
> Given that ROMAD requires only limited annotation overall, and bounding boxes remain the least expensive object-level annotation format, we believe our comparison is reasonable and not controversial.
>
> &nbsp;
>
> > **Q1: ROMAD’s slightly lower performance on LOCO**
>
> We do not believe that LOCO is “too simple” to benefit from relational modeling; LOCO contains several challenging logical anomalies. However, ROMAD achieves highly consistent performance across both datasets (94.48% AUROC on LOCO and 94.16% on CELAD), exhibiting the lowest performance variance among all baselines (Fig. 4). This generalization ability is a core strength of our approach.
>
> To be concrete, we aren't clear on what you mean by ROMAD slightly underperforming on LOCO, as ROMAD achieves better results on LOCO and performs equally well on both datasets. The only case where ROMAD underperforms is the “splicing connectors” class of LOCO, which has been the most challenging class of this dataset for all segmentation-based LAD methods.
>
> &nbsp;
>
> > **Q2: Sensitivity of distance-based attention (DBA) to object-density variations**
>
> We include a fine-grained comparison of ROMAD with and without the DBA module in the table below:
>
> | Dataset              | Without DBA | With DBA |
> |----------------------|-------------|----------|
> | breakfast box        | 95.96       | 94.54    |
> | juice bottle         | 92.71       | 94.04    |
> | pushpins             | 98.09       | 98.51    |
> | screw bag            | 98.82       | 98.50    |
> | splicing connectors  | 86.03       | 84.74    |
> | twin bracelets (CELAD)       | 89.56       | 94.08    |
>
>
> While some datasets—such as Breakfast Box, Pushpins, and Twin Bracelets—contain higher object density, we do not observe a strong correlation between object count and DBA-related performance gains or drops. The primary improvements from DBA occur on CELAD, whereas LOCO results remain nearly unchanged. We attribute this to the nature of LOCO’s logical constraints: most LOCO classes do not require precise spatial arrangement, making relative spatial proximity (the inductive bias introduced by DBA) less critical. In contrast, CELAD anomaly types rely more heavily on spatial and relational consistency.
>
> In summary, DBA is not sensitive to object density; rather, its effectiveness depends on the Type of relations required by the dataset’s logical constraints.

---

> ### Author Response · Authors · 2025-11-21
>
> > **Q3: Plans for extending CELAD**
>
>  1. **Additional object categories**
>
>     Yes; we originally intended to include more object categories, but collecting real-world LAD datasets is substantially more challenging than structural AD due to the need for novel logical constraints. We want to emphasise the fact that in LAD, simply adding more object categories does not necessarily increase task difficulty. In contrast to structural AD—where anomaly types share a common logic—LAD requires each category to present distinct logical constraints. Therefore, the diversity of anomaly types is more important than the diversity of object categories. CELAD introduces 31 logically distinct anomaly types—on average five times more than a LOCO class—representing a substantial expansion of logical complexity not previously covered in any LAD dataset.
>
>     Frankly, we have also experimented with additional object patterns but discarded them when they failed to introduce new logical constraints—models could solve them by exploiting local structural shortcuts rather than global relational understanding. For this reason, gathering real images offering genuinely new logical constraints is time-consuming and non-trivial.
>
>     In response to your question, we sought initial feedback from the field regarding whether our new approach would be beneficial for the community. Based on the experiences gained, we can always broaden the dataset by including more object categories.
>
>  2. **Extension to video anomaly detection**
>
>     We had not initially considered LAD for video anomaly detection, but we agree that the idea is promising. Temporal consistency could naturally introduce new forms of logical constraints. Exploring LAD in videos appears to be a compelling direction for future work and would require a dedicated follow-up study. The authors would greatly appreciate additional insights or suggestions on the recommended topic.

---

### Official Review · Reviewer_9hLU · 2025-11-01

**Soundness:** 2
**Presentation:** 2
**Contribution:** 2
**Rating:** 4
**Confidence:** 4

**Summary:**

This paper introduces CELAD, a new benchmark designed to test compositional logical anomaly detection, and proposes ROMAD (Relation-aware Object Matching for Anomaly Detection). ROMAD leverages a DETR-based detector to extract contextual object features, enriches them with distance-aware attention and area cues, and performs training-free bipartite matching against a memory bank of normal samples.  Experiments on MVTec LOCO AD and the new CELAD dataset show solid gains (≈ 14–15 %) and good few-shot performance.  While CELAD as a dataset is a meaningful step toward richer logical-composition evaluation, ROMAD itself mainly assembles existing components. Claims of logical or compositional reasoning are not well supported as shown form the experiments.

**Strengths:**

1. Well-timed attempt to move LAD research toward compositional evaluation.
2. The CELAD dataset appears carefully annotated and publicly released.
3. The proposed ROMAD model achieves strong few-shot anomaly detection for logical anomalies.
4. Experimental comparison with seven baselines (PSAD, CSAD, ComAD, SINBAD, EfficientAD, ULSAD, SA-PatchCore) shows superior performance of the proposed ROMAD method.

**Weaknesses:**

1. CELAD provides good benchmark contribution, but ROMAD showed limited methodological novelty. ROMAD offers little conceptual novelty, since it essentially combines DETR features with nearest-neighbor matching.
2. The proposed method heavily relies on the pretrained models; originality lies mainly in combination.
3. CELAD’s focus on bracelet-type compositions, but it provides limited generality.
4. The “compositional reasoning” claims exceed the presented evidence.
5. The experimental evaluation is performed on the logical anomalies in MVTec LOCO AD and CELAD datasets only. Including experimental results on more datasets, such as MVTec AD, would strengthen the paper by demonstrating broader applicability and robustness across different anomaly domains.

**Questions:**

1. The proposed ROMAD method was developed for detecting logical anomalies only. It seems to be quite restricted.

---

> ### Author Response · Authors · 2025-11-21
>
> Dear Reviewer 9hLU,
>
> Thank you for your constructive and insightful comments. We address each of your concerns in detail below. If any point requires further clarification, we would be happy to elaborate.
>
> > **W1 & W2: Lack of methodological novelty**
>
> ROMAD is intentionally designed to be simple, leveraging the relational modeling capabilities already present in DETR. We view this simplicity as a strength rather than a limitation. If a straightforward approach built on top of a foundation model can match or surpass heavily customized LAD architectures, this suggests that previous methods may not provide meaningful task-specific benefits and that the field may benefit from reevaluating its underlying assumptions.
>
> While ROMAD builds on existing pretrained components, the perspective we introduce—using DETR’s object query interactions as the basis for logical anomaly detection—is new. To the best of our knowledge, no prior work in LAD has explored DETR’s relational structure in this way. Moreover, we introduce an effective matching pipeline that adapts an established method in object detection into the specific requirements of LAD. Our novelty therefore lies not in architectural complexity, but in presenting a conceptually different and effective formulation of LAD.
>
> We believe this constitutes a valid methodological contribution, and that simplicity should not be mistaken for lack of novelty, especially when it leads to more generalizable and robust results.
>
> &nbsp;
>
> > **W3: Generality of CELAD**
>
> We interpret your argument as stating that “the dataset being limited to a single object category will provide limited generality.” We completely understand your concern but as stated in the paper, CELAD is an additional benchmark intended to complement the already existing datasets. We do not claim it to be a self-sufficient benchmark and we strongly encourage future works to evaluate across all available datasets to demonstrate maximum generalizability. In line with this principle, we have evaluated our method not only on CELAD but also on MVTec LOCO.
>
> Nevertheless, we want to emphasise the fact that in LAD, simply adding more object categories does not necessarily increase task difficulty. In contrast to structural AD—where anomaly types share a common logic—LAD requires each category to present distinct logical constraints. Therefore, the diversity of anomaly types is more important than the diversity of object categories. CELAD introduces 31 logically distinct anomaly types—on average five times more than a LOCO class—representing a substantial expansion of logical complexity not previously covered in any LAD dataset.
>
> &nbsp;
>
> > **W4: Strength of the “compositional reasoning” claim**
>
> We understand that the term “reasoning” is often used in the context of multi-step reasoning in LLMs. Our use of the term is more aligned with the established interpretation in object-centric vision models. Hence we aim to clarify our claims regarding the compositional understanding of ROMAD. As stated in our Method section:
>
> > “The contextualized object query embeddings encode positional, semantic, and compositional information of each object—respectively guided by the bounding box regression head, the classification head, and the self-attention mechanism.”
>
> Our claim is that DETR—and, by extension, ROMAD—captures positional, semantic, and compositional information of each object. We do not claim multi-step symbolic reasoning or high-level decision-making beyond these forms of relational representation.
>
> Nonetheless, taking into account  the architectural inductive biases built into DETR’s query design and explicit objectives to support each of the aforementioned aspects, we do not believe our claims regarding the compositional understanding to be an overstatement of the presented evidence.

---

> ### Author Response · Authors · 2025-11-21
>
> > **W5 & Q1: Why we did not focus on structural AD**
>
> Regarding the absence of structural AD results, our paper consistently motivates the importance of treating LAD as a distinct research direction. Our decision is based on several observations:
>
>  1. **Prior works already separate structural and logical AD into two independent components.**
>
>     Most existing methods—including PSAD, CSAD, GCAD, EfficientAD, PUAD, DSKD, and ULSAD—explicitly handle structural and logical anomalies through two separate branches. These branches are either completely independent or share only minimal components such as the image encoder. To highlight this more concretely:
>
>     - **PSAD** constructs its anomaly score from three memory banks. The “patch representation memory bank,” which is responsible for detecting fine-grained structural anomalies, is directly formed by using pixel-level scores of PatchCore, a well-established structural AD method.
>
>     - **CSAD** incorporates a dedicated “patch histogram branch” for logical anomalies and a separate “LGST branch” responsible for fine-grained structural anomalies. As stated in their paper, the LGST branch is a distillation-based structural AD module adapted from EfficientAD with only minor efficiency-related modifications.
>
>     These examples illustrate that current “unified” methods are typically rebranded ensembles of a logical module and a structural module, rather than genuinely integrated architectures. A truly unified model capable of jointly solving both tasks with competitive performance remains non-existent in the literature.
>
>  2. **LAD is the weaker link.**
>
>     Our dataset demonstrates that the logical anomaly types encountered in real-world scenarios are far more challenging than those previously covered, showcasing that current LAD approaches lag significantly behind structural AD methods. We therefore believe that focusing research effort on the more difficult and underexplored component—LAD—is both timely and impactful.
>
>     We also note that ComAD, an efficient and well-established method, is LAD-only, further indicating that LAD-focused methods are neither uncommon nor limited in applicability.
>
>  3. **Structural and logical anomalies require fundamentally different inductive biases.**
>
>     The nature of structural anomalies differs drastically from the relational understanding required for logical anomalies. This fundamental distinction explains why most current methods treat the two tasks as mixtures of separate models. It also suggests that attempting to unify them without addressing LAD-specific challenges first may be premature. Given that LAD is considerably more challenging than structural AD, a dedicated LAD method is no more restrictive in scope than existing structural-only approaches. Our work therefore concentrates on addressing what we view as the current bottleneck in AD research.
>
> Nonetheless, we verify compatibility with a structural branch. Following the aforementioned convention, we provide the results from ensembling our method with a structural branch (using RD4AD) on the MVTec AD dataset:
>
> | Dataset Class | Structural Branch | Logical Branch | Combined |
> |---------------|-------------------|----------------|----------|
> | Grid         | 99.7 | 78.5 | **99.7** |
> | Cable        | 96   | 68.4 | **96.2** |
> | Capsule      | 97.6 | 58.1 | **97.6** |
> | Bottle       | 99.9 | 67.7 | **100** |
> | Tile         | 99.9 | 88.2 | **99.9** |
> | Screw        | 98.7 | 54.2 | **98.5** |
> | Wood         | 99.4 | 87.7 | **99.5** |
> | Leather      | 100  | 84.2 | **100** |
> | Metal Nut    | 100  | 72.1 | **100** |
> | Zipper       | 98.6 | 91.1 | **99** |
> | Hazelnut     | 100  | 51.2 | **100** |
> | Transistor   | 96.9 | 92.6 | **99.8** |
> | Toothbrush   | 97.5 | 66.7 | **97.5** |
> | Pill         | 96.8 | 50.4 | **96.8** |
> | Carpet       | 98.6 | 91.9 | **99.2** |
>
> Please Note that the choice of RD4AD for our structural branch is not optimized for maximal performance due to the time constraints during the rebuttal phase. The results are just to demonstrate that our logical branch can be seamlessly combined with a structural branch, and the outputs of branches do not degrade one another.

---

### Official Review · Reviewer_iFrC · 2025-11-09

**Soundness:** 3
**Presentation:** 3
**Contribution:** 3
**Rating:** 8
**Confidence:** 5

**Summary:**

This paper introduces both a new pipeline for industrial (primarily logical) anomaly detection tasks (ROMAD), as well as a small dataset of logical bracelet-based anomalies (CELAD). ROMAD leverages a combination of DETR encoder-decoder to produce memory bank (during training) and anomaly candidates (during testing). These are then refined via SAM segmentation maps, producing corresponding features to both contrast against (as part of a separate memory bank), and used as queries (for test images). The resulting pipeline performs on state-of-the-art level in lower- to few-shot scenarios, and achieves highly competitive performance  in full-shot settings, tested on both the established MVTEC LOCO AD, and the newly introduced CELAD.

**Strengths:**

* The ROMAD pipeline is, to the best of my knowledge, a novel combination of memory-based AD methods combined with DETR & SAM detection plus segmentation for feature generation. The corresponding method pipeline seems very sensible. Given the pretraining of the associated modules and coupled with finetuning of the detection module, the resulting pipeline should be quite generally applicable.
* The corresponding performance on both MVTEC LOCO AD & their own CELAD benchmark is very convincing.
* CELAD itself is a novel, albeit rather small, contribution to the set of logical AD benchmarks & datasets.
* The paper itself is well written and structured.

**Weaknesses:**

The proposed method, and the corresponding benchmark, are two independent contributions. As CELAD itself is a simple additional benchmark for logical AD that does not exist yet, and the setup appears quite logical, I don't have any major issues to note there.
Regarding ROMAD, I have the following issues / questions:

* The detection module requires finetuning on manually annotated examples. It would be great if the authors could provide some context as to how the performance scales as a function of annotated examples, and how well it works out of the box without.
* Comparisons made in Table 1 includes methods that utilize components pretrained on different datasets and at different scales, s.a. e.g. EAD or SA-PC. It's always difficult to compare pipelines where technically, for competing methods, one could also consider swapping associated feature encoders with more modern or stronger ones. It would be great if the authors could provide additional information on how the numbers / relative differences should be best understood between different methods.
* How well can ROMAD be applied to non-logical AD benchmarks, e.g. plain MVTEC? It would be great to understand the limits of ROMAD as a general anomaly detection pipeline.

**Questions:**

See Weaknesses.

---

> ### Author Response · Authors · 2025-11-22
>
> Dear Reviewer iFrC,
>
> Thank you for your thoughtful and constructive feedback on our work. We appreciate your positive assessment of our contribution and the time you dedicated to reviewing the paper. Below, we provide a detailed response to your comments. If any point remains unclear, please let us know and we will gladly follow up.
>
> &nbsp;
>
> > **W1: ROMAD’s performance as a function of annotated samples**
>
> We find your observation both reasonable and important, as analyzing how ROMAD scales with the number of annotated samples provides deeper insight into its practical applicability. Below, we present additional experiments evaluating ROMAD’s performance as a function of the number of annotated samples. (Bold numbers correspond to the annotation budgets used in the main paper.)
>
>
> | Dataset class              | zero-shot | 2-shot | 4-shot | 6-shot | 8-shot | 10-shot |
> |----------------------------|-----------|--------|--------|--------|--------|---------|
> | screw bag                  | 50.44     | 97.31  | 98.61  | **99.14** | 99.22  | 99.20   |
> | splicing connectors        | 70.20     | 59.68  | 84.81  | **85.74** | 86.13  | 86.10   |
> | pushpins                   | 54.66     | 98.78  | 97.94  | **98.66** | 99.02  | 99.05   |
> | breakfast box              | 68.41     | 89.78  | 91.08  | **94.80** | 94.84  | 94.81   |
> | juice bottle               | 86.58     | 92.78  | 94.76  | **94.06** | 94.11  | 94.11   |
> | twin bracelets (CELAD)     | 51.37     | 71.86  | 76.38  | 93.01  | **94.16** | 94.31   |
>
> &nbsp;
>
> The table supports several conclusions:
>  - Due to having less object complexities, For LOCO classes, performance remains competitive even with fewer annotations. This aligns with our statement in the Experiments section:
>
>     > “The number of manually annotated samples for LOCO was not heavily optimized and could potentially be reduced even further.”
>
>  - For CELAD, performance decreases more sharply with fewer annotations. We attribute this to CELAD’s significantly higher object counts, larger number of object classes, and the presence of visually similar classes (e.g., the letters “P” and “R” could be mistaken by the detector if enough training samples are not provided).
>
>  - Each dataset class exhibits a saturation point beyond which additional annotations do not yield further improvements.
>
>  - ROMAD is not designed to operate without any annotations. Zero-shot performance is therefore outside the scope of the method, and—as expected—ROMAD does not perform well in such settings.
>
> &nbsp;
>
> > **W2: Baseline comparison details**
>
> For all baseline comparisons, we used the official codebases released by the authors. As you correctly noted, several baselines offer multiple architecture sizes. For instance, EfficientAD includes two variants: EfficientAD-S (the main architecture in the paper) and EfficientAD-M (a larger model with twice the number of hidden-layer kernels in both student and teacher networks).
>
> Since our comparisons focus on final performance rather than computational efficiency, we used the best-performing officially provided variant of each baseline (typically the largest). In the case of EfficientAD, this corresponds to EfficientAD-M. Importantly, we did not modify any baseline architecture by swapping backbones or introducing custom modifications, as this would deviate from standard practice. All selections were made strictly from officially mentioned model configurations.
>
> Moreover, fair baseline comparison involves additional considerations. For example, PSAD requires manual segmentation of samples, and the number of such samples, as well as the segmentation strategy, has significant impacts on performance. To account for all the details involved in the reproduction of each baseline, we have provided the complete codebases used for each baseline with updated environments, support for CELAD data loaders, and custom descriptions on how to easily run them in our provided anonymous repository:
> https://github.com/neutral-coder-737/Baselines
>
> Frankly, we dedicated significant time and effort to make sure our comparisons are reliable as much as possible. As noted in Appendix C:
> > “In total, considering both few- and full-shot settings, multiple datasets, and five independent runs per result, we conducted a **substantial number of 715 baseline runs**, representing a large-scale experimental effort to ensure the reliability of our comparisons. We are the first to evaluate such a broad set of models across so many settings.”

---

> ### Author Response · Authors · 2025-11-22
>
> > **W3: Why we did not focus on structural AD**
>
> Regarding the absence of structural AD results, our paper consistently motivates the importance of treating LAD as a distinct research direction. Our decision is based on several observations:
>
>  1. **Prior works already separate structural and logical AD into two independent components.**
>
>     Most existing methods—including PSAD, CSAD, GCAD, EfficientAD, PUAD, DSKD, and ULSAD—explicitly handle structural and logical anomalies through two separate branches. These branches are either completely independent or share only minimal components such as the image encoder. To highlight this more concretely:
>
>     - **PSAD** constructs its anomaly score from three memory banks. The “patch representation memory bank,” which is responsible for detecting fine-grained structural anomalies, is directly formed by using pixel-level scores of PatchCore, a well-established structural AD method.
>
>     - **CSAD** incorporates a dedicated “patch histogram branch” for logical anomalies and a separate “LGST branch” responsible for fine-grained structural anomalies. As stated in their paper, the LGST branch is a distillation-based structural AD module adapted from EfficientAD with only minor efficiency-related modifications.
>
>     These examples illustrate that current “unified” methods are typically rebranded ensembles of a logical module and a structural module, rather than genuinely integrated architectures. A truly unified model capable of jointly solving both tasks with competitive performance remains non-existent in the literature.
>
>  2. **LAD is the weaker link.**
>
>     Our dataset demonstrates that the logical anomaly types encountered in real-world scenarios are far more challenging than those previously covered, showcasing that current LAD approaches lag significantly behind structural AD methods. We therefore believe that focusing research effort on the more difficult and underexplored component—LAD—is both timely and impactful.
>
>     We also note that ComAD, an efficient and well-established method, is LAD-only, further indicating that LAD-focused methods are neither uncommon nor limited in applicability.
>
>  3. **Structural and logical anomalies require fundamentally different inductive biases.**
>
>     The nature of structural anomalies differs drastically from the relational understanding required for logical anomalies. This fundamental distinction explains why most current methods treat the two tasks as mixtures of separate models. It also suggests that attempting to unify them without addressing LAD-specific challenges first may be premature. Given that LAD is considerably more challenging than structural AD, a dedicated LAD method is no more restrictive in scope than existing structural-only approaches. Our work therefore concentrates on addressing what we view as the current bottleneck in AD research.
>
> Nonetheless, we verify compatibility with a structural branch. Following the aforementioned convention, we provide the results from ensembling our method with a structural branch (using RD4AD) on the MVTec AD dataset:
>
> | Dataset Class | Structural Branch | Logical Branch | Combined |
> |---------------|-------------------|----------------|----------|
> | Grid         | 99.7 | 78.5 | **99.7** |
> | Cable        | 96   | 68.4 | **96.2** |
> | Capsule      | 97.6 | 58.1 | **97.6** |
> | Bottle       | 99.9 | 67.7 | **100** |
> | Tile         | 99.9 | 88.2 | **99.9** |
> | Screw        | 98.7 | 54.2 | **98.5** |
> | Wood         | 99.4 | 87.7 | **99.5** |
> | Leather      | 100  | 84.2 | **100** |
> | Metal Nut    | 100  | 72.1 | **100** |
> | Zipper       | 98.6 | 91.1 | **99** |
> | Hazelnut     | 100  | 51.2 | **100** |
> | Transistor   | 96.9 | 92.6 | **99.8** |
> | Toothbrush   | 97.5 | 66.7 | **97.5** |
> | Pill         | 96.8 | 50.4 | **96.8** |
> | Carpet       | 98.6 | 91.9 | **99.2** |
>
> Please Note that the choice of RD4AD for our structural branch is not optimized for maximal performance due to the time constraints during the rebuttal phase. The results are just to demonstrate that our logical branch can be seamlessly combined with a structural branch, and the outputs of branches do not degrade one another.

---

### Meta-Review · Area_Chair_ekQm · 2025-12-04

**Summary:**

This paper introduces a logic-anomaly benchmark based on wristband-triggered events. However, as noted by Reviewers 4exq, jtKz, and 9hLU, the benchmark covers only a single type of logical anomaly, resulting in limited data diversity. For a top-tier venue like ICLR, the contribution is not sufficiently rich or comprehensive. Therefore, I consider the paper unsuitable for acceptance.

**Reviewer Concerns:**

The authors addressed the concern regarding structural anomalies by incorporating combined evaluations with structural anomaly detection.  But concerns about broader logical anomalies, and more evaluations are not addressed.

**Reviewer Scores:**

It is unlikely that reviewers raise their scores. Aside from adding one supplementary experiment on structural anomalies, the authors did not provide substantial additional evidence in response to the negative reviewers (Reviewers 4exq, jtKz, and 9hLU). Their concerns—primarily regarding the limited coverage beyond wristband-like cases and insufficient performance in broader scenarios—remain largely unaddressed.

---

### Decision · Program_Chairs · 2026-01-26

Reject